# Serum and Plasma Tumor Necrosis Factor Alpha Levels in Individuals with Obstructive Sleep Apnea Syndrome: A Meta-Analysis and Meta-Regression

**DOI:** 10.3390/life10060087

**Published:** 2020-06-12

**Authors:** Mohammad Moslem Imani, Masoud Sadeghi, Habibolah Khazaie, Mehrdad Emami, Dena Sadeghi Bahmani, Serge Brand

**Affiliations:** 1Department of Orthodontics, Kermanshah University of Medical Sciences, 6713954658 Kermanshah, Iran; mmoslem.imani@yahoo.com; 2Medical Biology Research Center, Kermanshah University of Medical Sciences, 6714415185 Kermanshah, Iran; sadeghi_mbrc@yahoo.com; 3Sleep Disorders Research Center, Kermanshah University of Medical Sciences, 6719851115 Kermanshah, Iran; hakhazaie@gmail.com (H.K.); dena.sadeghibahmani@upk.ch (D.S.B.); 4Students Research Committee, Kermanshah University of Medical Sciences, 6715847141 Kermanshah, Iran; master_com63@yahoo.com; 5Center for Affective, Stress and Sleep Disorders, University of Basel, Psychiatric Clinics, 4001 Basel, Switzerland; 6Substance Abuse Prevention Research Center, Kermanshah University of Medical Sciences, 6715847141 Kermanshah, Iran; 7Departments of Physical Therapy, University of Alabama at Birmingham, Birmingham, AL 35209, USA; 8Department of Sport, Exercise and Health, Division of Sport Science and Psychosocial Health, University of Basel, 4052 Basel, Switzerland; 9School of Medicine, Tehran University of Medical Sciences, 1416753955 Tehran, Iran

**Keywords:** obstructive sleep apnea syndrome, cytokine, tumor necrosis factor alpha, serum, plasma, meta-analysis, meta-regression

## Abstract

Background: Obstructive sleep apnea syndrome (OSAS) is associated with a variety of inflammatory factors. Specifically, proinflammatory cytokines appear to be associated with the pathogenesis of OSAS. Methods: For the present meta-analysis and meta-regression on serum and plasma tumor necrosis factor alpha (TNF-α) levels in individuals with and without OSAS, we performed a systematic search without any restrictions of the PubMed/Medline, Scopus, Cochrane Library, and Web of Science databases to find relevant articles published up to 1 February 2020. Results: Fifty-five (adults: 29 studies on serum and 17 studies on plasma; children: 4 studies on serum and 5 studies on plasma) were included and analyzed. Always compared to age-matched healthy controls, the pooled MDs were as follows: adults, serum: 10.22 pg/mL (95% CI = 8.86, 11.58; *p* < 0.00001); adults, plasma: 5.90 pg/mL (95% CI = 4.00, 7.80; *p* < 0.00001); children, serum: 0.21 pg/mL (95% CI = 0.05, 0.37; *p* = 0.01); children, plasma: 5.90 pg/mL (95% CI = 4.00, 7.80; *p* < 0.00001). Conclusions: Compared to healthy and age-matched controls, adult individuals with OSAS had significantly higher serum/plasma TNF-α levels. For children with OSAS, significantly higher levels were observed for TNF-α in serum but not in plasma.

## 1. Introduction

Obstructive sleep apnea syndrome (OSAS) is a common sleep-related breathing disorder. It is estimated that 4% of both female and male adults suffer from OSAS [1]. Specifically, about 2% middle-aged individuals (30–60 years) suffer from OSAS, and prevalence rates increased by 30% to 50% in adults with overweight and obesity [2,3]. Typically, in individuals with OSAS, the repetitive obstruction of the upper airway during sleep leads to hypoxemia, arousals, intermittent snoring, episodes of complete apnea, and to increased daytime sleepiness as a result of non-restoring sleep [4]. To define OSAS, both objective and subjective measurements are applied: While self-rating questionnaires such as the STOP-Bang [5,6] allow a rough screening, polysomnography is the gold standard: OSAS are defined as the Apnea–Hypopnea Index (AHI) > 5 events per hour [7].

People with objectively assessed OSAS are at increased risk to suffer from atherogenesis [8], vascular dysfunction [9], congestive heart failure, hypertension, atrial fibrillation, nocturnal arrhythmias, stroke, pulmonary hypertension, and metabolic syndrome [10,11].

As regards the immune system, the tumor necrosis factor alpha (TNF-α) is a common cytokine secreted by adipocytes and the mononuclear-macrophage system [12], which regulates the immune system, induces inflammation, and participates in the regulation of fat metabolism [13]. More specifically, several inflammatory cells secrete proinflammatory cytokines such as TNF-α. Increased TNF-α concentrations can cause inflammatory responses such as neutrophil recruitment, tissue destruction, neovascularization [14], oxidative stress, systemic inflammation, autonomic nervous system dysfunction, atherosclerosis, and aging [15]. Further, highly increased TNF-α concentrations appear to be causally associated with the pathogenesis of diseases such as metabolic diseases, cardiovascular diseases, and OSAS [16]. Further, higher TNF-α concentrations were observed in individuals with OSAS, compared to healthy controls. In 2013, Nadeem et al. [17] showed in their meta-analysis of 15 case-control studies that compared to controls, individuals with OSAS had significantly higher TNF-α concentrations. Similarly, in 2015, Wang et al. [18] also reported in their meta-analysis that compared to healthy controls, individuals with OSAS had significantly higher serum TNF-α concentrations. With the present meta-analysis, we expand upon the previous two meta-analyses in that we focused on (a) both on serum and on plasma TNF-α, (b) on samples with adults and children with OSAS, and in that (c) we performed a meta-regression, and (d) we ran sub-group analyses.

## 2. Materials and Methods

The meta-analysis was performed in accordance with the Preferred Reporting Items for Systematic Reviews and Meta-Analyses (PRISMA) guidelines [19].

### 2.1. Search Strategy

One of the authors (M. Sadeghi) systematically searched four databases, namely PubMed/Medline, Scopus, Cochrane Library, and Web of Science, for articles published in English up to 1 February 2020, with no restrictions, and with the following expressions: “sleep apnea” or “OSA” or “obstructive sleep apnea” or “obstructive sleep apnea syndrome” or “OSAS“, with “TNF-alpha” or “TNF-α” or “TNF” or “tumor necrosis factor alpha”; and with “blood” or “plasma” or “serum”. We also manually inspected the reference lists of the articles (original and review articles) for publications related to our topic.

### 2.2. Eligibility Criteria

Inclusion criteria were as follows: (1) studies evaluating the association between plasma or serum TNF-α levels and OSAS with case-control design without age (≥18 years as adults and <18 years as children), sex or BMI restrictions; (2) OSAS was defined as AHI > 5 events/h diagnosed by polysomnography; (3) participants with OSAS and controls had no other systematic diseases such as diabetes, asthma, neurological disorders such as multiple sclerosis, neurodegenerative disorders such as Alzheimer’s disease, or oral diseases; (4) studies reporting pretreatment morning serum and/or plasma levels of TNF-α (around 6:00–10:00 a.m.); (5) studies reporting sufficient data to compute the MD and 95% confidence interval (CI) in participants with OSAS and controls; (6) studies with more than 10 cases included as individuals with OSAS and controls.

Exclusion criteria were: (1) studies with irrelevant or insufficient data or without clinical data; (2) Letter to the editors, review articles, conference papers, meeting abstracts, and book chapters; (3) studies without a control group; (4) studies reporting controls with AHI > 5 events/h; (5) studies with overlapped data with other studies; (6) studies without reporting cut-off AHI values for OSAS; (7) studies mixing pediatric and adult participants.

### 2.3. Study Selection

Two authors (M.M.I. and M.E.) read independently the titles and abstracts of the retrieved studies. They then selected the relevant studies, while another author (M.S.) retrieved the full-texts of the articles and excluded several full-texts based on the exclusion criteria mentioned above. If two studies had overlapped data, we selected the study with the most recent publication year.

### 2.4. Data Extraction

Two authors (S.B. and M.S.) independently extracted the data from each study included in the meta-analysis. If there was a disagreement between the two authors, the third author (M.M.I.) helped to find a final decision. The data extracted for the meta-analysis included basic information including the first author, publication year, country of study, ethnicity of subjects in each study, age, BMI, and AHI of both groups, and mean and standard deviation (SD) of TNF-α levels in plasma and serum. Since some studies considered patients with mild OSAS (AHI: 1–5 events/h) as a control group condition, and since we had to use this protocol for other studies, reporting the quality of studies as for usual case-control studies did not seem to be reasonable.

### 2.5. Quality Assessment

One author (M.S) evaluated the quality of the studies included in the meta-analysis using the Newcastle-Ottawa Scale (NOS) with a total score of nine for each study [20].

### 2.6. Statistical Analyses

The data were analyzed by one author. Review Manager 5.3 software (RevMan 5.3, (RevMan 5.3, The Cochrane Collaboration, Oxford, UK) was used to calculate the crude MD and 95% CI, which evaluated the significance of the pooled MD by Z-test. Heterogeneity across the studies was evaluated using both Cochran Q [21] and I2 metrics with scores ranging from 0% to 100% [22]. In addition, when *p*-value < 0.1 and I2 > 50% showed a statistically significant heterogeneity, the analysis was performed by the random-effect model to evaluate the pooled ORs and CI values. Otherwise, we used the fixed-effect model. The weight (%) represents the effect of an individual study on the pooled results obtained. In general, the larger the sample size and the narrower CI are, the higher is the percentage of weight.

The results of the Begg’s and Egger’s tests were analyzed by the Comprehensive Meta-Analysis version 2.0 software (CMA 2.0). The Begg’s funnel plot shows the standard error (SE) of the log (MD), and the precision of each study is plotted against its log (MD) [23]. In addition, the Egger’s test shows the linear regression between the precision of the studies and the standardized effect [24].

Subgroup analyses were performed based on ethnicity, AHI, BMI, and number of participants. The sensitivity analyses, namely the “cumulative analysis” and “one study removed”, were applied to estimate the consistency/stability of the results. If the *p*-value (two-tailed) was less than 0.05, then there was a statistically significant difference. The meta-regression is a quantitative method used in meta-analyses to estimate the impact of moderator variables on study effect size. The trim-and-fill method was used to estimate potentially missing studies due to publication bias in the funnel plot and adjusting the overall effect estimate [25].

Some studies reported the values of TNF-α in standard errors (SE); SEs were transformed into standard deviation (SD) (SE = SD/√*N*; *N* = number of individuals). Some studies reported median and interquartile values, which were transformed into mean and SD [26]. The serum and plasma TNF-α levels were reported in picograms per milliliter (pg/mL). Participants with BMI >30 kg/m^2^ were considered obese [27].

## 3. Results

### 3.1. Search Strategies

The flow chart (Figure 1) sets out the selection process. The search databases yielded 912 articles; of these, 117 full-texts met eligibility criteria after removing duplicates and irrelevant studies. Next, 62 articles were removed with reasons (two were animal studies, three were meta-analyses, six reported polymorphisms of TNF-alpha, twenty-seven either had no control or control group was selected as AHI > 5, five were reviews, two had less than ten cases included in both groups, one reported high sensitivity TNF alpha, four had no sufficient data, one had duplicate data with another study, three showed production of TNF alpha by cells, two did not show cut-off of AHI for OSAS, one was the letter to editor, one mixed adults and pediatric samples, one had controls with recurrent tonsillar infection, one had controls with bronchial asthma, one had controls with diabetes, one had no relevant data). This left 55 studies to be analyzed in the meta-analysis [14,16,28,29,30,31,32,33,34,35,36,37,38,39,40,41,42,43,44,45,46,47,48,49,50,51,52,53,54,55,56,57,58,59,60,61,62,63,64,65,66,67,68,69,70,71,72,73,74,75,76,77,78,79,80].

### 3.2. Characteristics of Studies

Table 1 presents the characteristics of the 55 studies included in the meta-analysis. Forty-six studies [16,28,29,30,31,32,33,34,35,36,38,39,40,41,43,44,45,46,47,48,49,50,51,52,54,56,57,58,59,60,61,62,63,64,65,67,68,69,70,71,72,75,76,78,79,80] reported serum or plasma TNF-α concentrations in adults, and nine studies [14,37,42,53,55,66,73,74,77] reported serum or plasma TNF-α concentrations in children and adolescents. Among adult participants, eighteen studies were performed with Caucasians [31,34,36,39,40,41,46,47,48,49,59,61,63,65,68,69,75,80], nineteen sampled Asians [16,29,32,33,35,43,45,52,54,58,60,62,64,67,71,72,76,78,79], and nine samples mixed [28,30,38,44,50,51,56,57,70] ethnicities. For studies with adult samples, 29 out of 46 studies reported serum TNF-α concentrations [16,31,32,33,34,35,36,38,40,41,45,46,48,50,51,52,56,60,61,63,64,65,67,68,70,76,78,79,80] and 17 reported plasma TNF-α concentrations [28,29,30,39,43,44,47,49,54,57,58,59,62,69,71,72,75].

### 3.3. Overall Analysis

#### 3.3.1. Serum Levels of TNF-α in Adults

Twenty-nine studies compared serum TNF-α concentrations between adults with OSAS and healthy controls (Figure 2); adults with OSAS had significantly higher TNF-α concentrations compared to healthy controls: 10.22 pg/mL (95% CI = 8.86, 11.58; *p* < 0.00001; I^2^ = 100% (P_h_ or P_heterogeneity_ < 0.00001)).

#### 3.3.2. Plasma Levels of TNF-α in Adults

Seventeen studies compared plasma TNF-α concentrations between adults with OSAS and healthy controls (Figure 3) and found significantly higher TNF-α concentrations in those with OSAS: 5.90 pg/mL (95% CI = 4.00, 7.80; *p* < 0.00001; I^2^ = 99% (P_h_ < 0.00001)).

#### 3.3.3. Serum TNF-α Levels in Children

Four studies compared serum TNF-α concentrations between children with OSAS and without OSAS (Figure 4) and found significantly higher TNF-α concentrations in those with OSAS: 0.21 pg/mL (95% CI = 0.05, 0.37; *p* = 0.01; I^2^ = 77% [P_h_ = 0.005].

#### 3.3.4. Plasma TNF-α Levels in Children

Five studies compared serum TNF-α concentrations between children with OSAS and without OSAS (Figure 5) and found no statistically significant MDs: 0.15 pg/mL (95% CI = −0.16, 0.45; *p* = 0.35; I^2^ = 61% [P_h_ = 0.04]).

### 3.4. Overall Subgroup Analysis of Serum and Plasma TNF-α Levels

#### 3.4.1. Ethnicity

Subgroup analysis of serum TNF-α concentrations in adult samples are reported in Table 2. The pooled analysis showed that for those with OSAS, serum TNF-α concentrations in Asians (MD = 37.11 pg/mL (95% CI = 33.27, 40.95; *p* < 0.00001), in Caucasians (MD = 2.20 pg/mL; 95% CI = 1.06, 3.34; *p* = 0.0002), and in mixed ethnicities (MD = 0.43; 95% CI = 0.30, 0.56; *p* < 0.00001) were significantly higher, compared to healthy controls. Serum TNF-α concentrations in Asians were 16.9 and 86.3 times higher than serum TNF-α concentrations in Caucasian and mixed ethnicities with OSAS, and always compared to controls.

For plasma TNF-α concentrations, for those adults with OSAS, plasma TNF-α concentrations in Asians (MD = 14.92 pg/mL; 95% CI = 9.86, 19.98; *p* < 0.00001), in Caucasians (MD = 3.72 (95% CI = 1.04, 6.40; *p* = 0.007)), and in mixed ethnicities (MD = 1.22; 95% CI = 0.10, 2.34; *p* = 0.03) were significantly higher, compared to healthy controls. Next, in Asians, plasma TNF-α concentrations were 4 and 12.2 times higher than in Caucasian and mixed ethnicities with OSAS, and always compared to controls.

#### 3.4.2. Mean BMI of Individuals with OSAS

For serum TNF-α concentrations, and compared to healthy controls, adult participants with OSAS and a BMI > 30 km/m^2^ showed a pooled MD of 1.27 (95% CI: 0.67, 1.86; *p* < 0.0001). For adult participants with OSAS and a BMI ≤ 30 km/m^2^, the pooled MD was 23.07 pg/mL (95% CI: 20.18, 25.96; *p* < 0.00001). Thus, results showed that serum TNF-α concentrations were 18.2 times higher participants with OSAS and a BMI > 30 kg/m^2^, when compared to those participants with a mean BMI ≤30 kg/m^2^.

For plasma TNF-α concentrations, and compared to healthy controls, adult participants with OSAS and a BMI > 30 km/m^2^ showed a pooled MD of 1.06 pg/mL (95% CI: 0.34, 1.78; *p* = 0.004). For adult participants with OSAS and a BMI ≤ 30 km/m^2^, the pooled MD was 10.82 pg/mL (95% CI: 7.14, 14.49; *p* < 0.00001). Thus, results showed that plasma TNF-α concentrations were 10.2 times higher in adults with OSAS and a BMI > 30 kg/m^2^ than plasma TNF-α concentrations of adults with OSAS and a BMI ≤ 30 kg/m^2^ and always compared to healthy controls.

#### 3.4.3. Mean BMI of Controls

For serum TNF-α levels, compared to controls, individuals with OSAS and with a BMI > 30 kg/m^2^ showed a pooled MD of 2.28 pg/mL (95% CI: 0.86, 3.70; *p* = 0.002); while compared to controls, individuals with OSAS and with a BMI ≤ 30 kg/m^2^ showed a pooled MD of 18.64 pg/mL (95% CI: 16.33, 20.95; *p* < 0.00001). Thus, serum TNF-α levels were 8.2 times higher in studies with participants with OSAS and a BMI >30 kg/m^2^ than serum TNF-α levels with participants with OSAS and a BMI ≤ 30 kg/m^2^.

For plasma TNF-α levels, compared to controls, the pooled MD was 7.32 pg/mL (95% CI: 5.02, 9.62; *p* < 0.00001) for participants with OSAS and a BMI > 30 kg/m^2^; for participants with OSAS and a BMI ≤ 30 kg/m^2^, the pooled MD was not statistically significant (MD = 1.19 pg/mL; 95% CI: 0.00, 2.38; *p* = 0.05).

#### 3.4.4. Total Number of Participants

For serum TNF-α levels, the pooled MD was 55.47 pg/mL (95% CI: 47.21, 63.72; *p* < 0.00001) for studies with more than 100 cases, and 2.15 pg/mL (95% CI: 1.57, 2.72; *p* < 0.00001) for studies with less than 100 cases. Thus, serum TNF-α levels were 25.8 times higher in the studies with more 100 cases, compared to studies with less than 100 cases.

For plasma TNF-α levels, the pooled MD was 6.91 pg/mL (95%CI: 2.20, 11.63; *p* = 0.004) for studies with more than 100 cases, while the pooled MD was 2.87 pg/mL (95%CI: 1.37, 4.37; *p* = 0.0002) for studies with less than 100 cases. Thus, plasma TNF-α levels were 2.4 times higher in the studies with more than 100 cases, compared to studies with less than 100 cases.

#### 3.4.5. Mean AHI of Individuals with OSAS

For serum TNF-α levels, in individuals with OSAS and a AHI > 30 events/h, the pooled MD was 5.72 pg/mL (95% CI: 4.32, 7.12; *p* < 0.00001), while the pooled MD was 127.01 pg/mL (95%CI: 100.56, 153.47; *p* < 0.00001) in individuals with OSAS and a AHI ≤ 30 events/h. Thus, serum TNF-α levels were 22.2 times higher, when participants with OSAS had AHI ≤ 30 events/h, compared to AHI > 30 events/h.

For plasma TNF-α levels, in individuals with OSAS and AHI > 30 events/h, the pooled MD was 8.15 pg/mL (95% CI: 5.59, 10.71; *p* < 0.00001), while in individuals with OSAS and AHI ≤ 30 events/h, the pooled MD was 2.97 pg/mL (95% CI: 0.21, 5.73; *p* = 0.03). Thus, serum TNF-α levels were 2.7 times higher in participants with OSAS and AHI > 30 events/h, compared to serum TNF-α levels in participants with OSAS and AHI ≤ 30 events/h.

### 3.5. Adult Caucasian Participants

#### 3.5.1. Mean BMI of Individuals with OSAS

For serum TNF-α levels, studies with Caucasian participants with OSAS and a BMI > 30 kg/m^2^ showed the pooled MD of 2.58 pg/mL (95% CI: 1.07, 4.10; *p* = 0.0008), whereas for studies with participants with a BMI ≤ 30 kg/m^2^ serum TNF-α levels did not statistically differ between participants with or without OSAS (Table 3).

For plasma TNF-α levels, studies with Caucasian participants with OSAS and a BMI ≤ 30 kg/m^2^ showed the pooled MD of 7.99 pg/mL (95% CI: 1.84, 14.15; *p* = 0.01), whereas no significant MDs were observed in studies including Caucasian participants with OSAS and a BMI > 30 kg/m^2^.

#### 3.5.2. Mean BMI of Controls

For studies with serum TNF-α levels and Caucasian controls with a mean BMI > 30 kg/m^2^, the pooled MD was 2.58 pg/mL (95% CI: 1.07, 4.10; *p* = 0.0008). There was no statistically significant MD for serum TNF-α levels between Caucasians with or without OSAS and a mean ≤ 30 kg/m^2^.

For studies with plasma TNF-α levels and controls with a BMI > 30 kg/m^2^, the pooled MD was 0.56 pg/mL (95% CI: 0.05, 1.08; *p* = 0.03). Studies with controls with a BMI ≤ 30 kg/m^2^ showed a pooled MD of 5.64 pg/mL (95% CI: 0.77, 10.52; 0.02). Thus, plasma TNF-α levels were 10.1 times higher in controls with a BMI ≤ 30 kg/m^2^, compared to plasma TNF-α levels of controls with a BMI > 30 kg/m^2^

#### 3.5.3. Total Number of Participants

Serum TNF-α levels had a significant difference in studies with ≤ 100 cases (MD = 3.21 pg/mL; 95% CI: 1.37, 5.05; *p* = 0.0006), whereas there was no significant difference for studies with >100 cases.

#### 3.5.4. Mean AHI of Individuals with OSAS

In studies with individuals with OSAS and AHI of > 30 events/h, serum TNF-α levels were 1.40 pg/mL (95% CI: 0.34, 2.46; *p* = 0.010), and plasma TNF-α levels were 7.80 pg/mL (95% CI: 2.47, 13.13; *p* = 0.004).

### 3.6. Adult Asian Participants

#### 3.6.1. Mean BMI of Individuals with OSAS

For serum and plasma TNF-α levels, there were no Asian participants with OSAS and a mean BMI > 30 kg/m^2^. For studies with Asian participants with OSAS and a mean BMI ≤ 30 kg/m^2^, serum (MD = 37.11 pg/mL; 95% CI: 33.27, 25.96; *p* < 0.0001) and plasma (MD = 14.92 pg/mL; 95% CI: 9.86, 19.98; *p* < 0.00001) TNF-α levels differed statistically significantly (Table 4).

#### 3.6.2. Mean BMI of Controls

For serum and plasma TNF-α levels, there were no studies including controls with a BMI > 30 kg/m^2^. The serum TNF-α levels were 37.11 pg/mL (95% CI: 33.27, 25.96; *p* < 0.0001) for studies including OSAS patients with a mean BMI of ≤30 kg/m^2^, and plasma TNF-α levels were 14.92 pg/mL (CI: 9.86, 19.98; *p* < 0.00001) for studies including OSAS patients with a mean BMI of ≤30 kg/m^2^.

#### 3.6.3. Total Number of Participants

For serum TNF-α levels, the pooled MD was 89.88 pg/mL (95% CI: 74.59, 105.17; *p* < 0.00001) for studies with 100 or more participants, and the pooled MD was 7.80 pg/mL (95% CI: 5.76, 9.85; *p* < 0.00001) for studies less than 100 participants. Thus, serum TNF-α levels were 11.5 times higher in studies with more than 100 participants, compared to studies with less than 100 participants.

For plasma TNF-α levels, the pooled MD was 12.34 pg/mL (95%CI: 7.84, 16.84; *p* < 0.00001) for studies with 100 or more participants, and the pooled MD was 21.51 pg/mL (95%CI: 13.15, 29.87; *p* < 0.00001) for studies less than 100 participants. Thus, plasma TNF-α levels were 1.7 times higher in studies with less than 100 participants, compared to studies with more than 100 participants.

#### 3.6.4. Mean AHI of Individuals with OSAS

No study investigated serum and plasma TNF-α levels in participants with OSAS with a mean AHI ≤ 30 events/h.

For studies including AHI > 30 events/h, mean serum TNF-α levels were 12.94 (pg/mL; 95% CI: 7.74, 18.13; *p* < 0.00001); plasma TNF-α levels were 12.48 pg/mL (95% CI: 17.04, 31.52; *p* < 0.00001).

### 3.7. Adult Participants Including Mean BMI of Individuals with OSAS > 30 kg/m^2^

#### 3.7.1. Mean BMI of Controls

For studies with serum TNF-α levels and controls with BMI of controls >30 kg/m^2^, there was a significant difference for serum TNF-α levels (MD = 2.28 pg/mL; 95% CI: 0.86, 3.70; *p* = 0.002), compared to studies with controls with BMI of controls ≤ 30 kg/m^2^ (Table 5).

#### 3.7.2. Total Number of Participants

For studies with 100 or less participants, there was a significant difference for serum TNF-α levels (MD = 1.27 pg/mL; 95% CI: 0.65, 1.89; *p* < 0.0001) and for plasma TNF-α levels (MD = 1.51 pg/mL; 95% CI: 0.53, 2.49; *p* = 0.003), compared to studies with more than 100 participants.

#### 3.7.3. Mean AHI of Participants with OSAS

For serum TNF-α levels, the pooled MD for studies including participants with OSAS with AHI > 30 events/h was 0.96 pg/mL (95% CI: 0.50, 1.42; *p* < 0.0001); the pooled MD for studies with participants with OSAS with AHI ≤ 30 events/h was 42.13 pg/mL (95%CI: 33.48, 50.77; *p* < 0.00001). Thus, serum TNF-α levels were 42.9 times higher in studies with participants with OSAS and AHI ≤ 30 events/h, compared to the serum TNF-α levels in studies including participants with OSAS and AHI > 30 events/h.

For plasma TNF-α levels, the pooled MD for studies with participants with OSAS with AHI > 30 events/h was 1.69 pg/mL (95% CI: 0.47, 2.90; *p* = 0.007). When studies included participants with OSAS and an AHI ≤ 30 events/h, MDs between studies with participants with or without OSAS were not significant.

### 3.8. Adult Participants with a Mean BMI ≤ 30 kg/m^2^ in Participants with OSAS

#### 3.8.1. Mean BMI of Controls

No study investigated serum and plasma TNF-α levels in participants with OSAS with a mean BMI >30 kg/m^2^.

For studies including BMI ≤ 30 kg/m^2^ in controls, TNF-α levels were 23.07 pg/mL (95% CI: 20.18, 4.50; *p* < 0.00001) for serum, and MD = 10.82 pg/mL (95% CI: 7.14, 14.49; *p* < 0.00001) for plasma (Table 6).

#### 3.8.2. Total Number of Participants

For serum TNF-α levels and in studies with more than 100 participants, the pooled MD was 63.51 pg/mL (95% CI: 53.65, 73.45; *p* < 0.00001), while in studies with ≤ 100 participants, the pooled MD was 4.42 pg/mL (95% CI: 3.10, 5.74; *p* < 0.00001). Thus, serum TNF-α levels were 14.4 times higher in studies with more than 100 participants, compared to serum TNF-α levels in studies with ≤100 participants.

For plasma TNF-α levels and in studies with more than 100 participants, the pooled MD was 8.12 pg/mL (95%CI: 4.46, 11.79; *p* < 0.0001) while in studies with ≤100 participants, the pooled MD 13.79 pg/mL (95%CI: 8.24, 19.35; *p* < 0.00001). Thus, plasma TNF-α levels were 1.7 times higher in studies with more than 100 participants, compared to plasma TNF-α levels in studies with ≤100 participants.

#### 3.8.3. Mean AHI of Participants with OSAS

For serum TNF-α levels, the pooled MD for studies including participants with OSAS and an AHI > 30 events/h was 12.94 pg/mL (95% CI: 7.74, 18.13; *p* < 0.00001); the pooled MD for studies including participants with OSAS and an AHI ≤ 30 events/h was 173.00 pg/mL (95%CI: 139.66, 206.34; *p* < 0.00001). Thus, serum TNF-α levels were 13.4 times higher in participants with OSAS and AHI ≤ 30 events/h, compared to the serum TNF-α levels in participants with OSAS and AHI > 30 events/h.

For plasma TNF-α levels, the pooled MD for studies including participants with OSAS and an AHI > 30 events/h was 16.13 pg/mL (95% CI: 11.20, 21.06; *p* < 0.00001). When studies included participants with OSAS had an AHI ≤ 30 events/h, MDs between studies including participants with or without OSAS were not significant.

### 3.9. Meta-Regression

The results of meta-regression showed that age, the publication year, the mean BMI, the mean AHI, and the number of participants did not systematically change serum and plasma TNF-α levels (Table 7).

### 3.10. Quality Assessment

The quality score of each study included in the meta-analysis is illustrated in Table 8.

### 3.11. Sensitivity Analysis

The “cumulative analysis” and the “one study removed” as two sensitivity analyses showed the stability of the results. In addition, excluding statistically the studies with outliers data did not change pooled analysis of serum (MD = 4.53 pg/mL, *p* < 0.00001) and plasma (MD = 4.18 pg/mL, *p* < 0.00001) TNF-α levels (Table 9).

### 3.12. Publication Bias

Figure 6 shows the funnel plots of the analysis of serum and plasma TNF-α levels and Table 10 illustrates the results of trim-and-fill method on bias.

For serum levels, Egger’s and Begg’s tests (*p* = 0.02431 and *p* = 0.04677, respectively) revealed a publication bias. For plasma TNF-α levels, Begg’s test (*p* = 0.03943) revealed a publication bias, but Egger’s test (*p* = 0.19315) did not show any bias between and across the studies.

For serum TNF-α levels and 15 imputed studies, under the fixed-effects model, the point estimate and pseudo 95% CI for the combined studies was 0.410 (0.365, 0.455); using the trim-fill method, the imputed point estimate was 0.323 (0.278, 0.368). In addition, under the random-effects model, the point estimate and 95% CI for the combined studies was 8.245 (7.006, 9.483); using the trim-fill method, the imputed point estimate was 0.523 (−0.889, 1.935).

For plasma TNF-α levels and 3 imputed studies, under the fixed-effects model, the point estimate and 95% CI for the combined studies was 2.463 (2.323, 2.602), and using trim-fill method, the imputed point estimate was 2.294 (2.156, 2.433). In addition, under the random-effects model, the point estimate and pseudo 95% CI for the combined studies was 5.452 (3.501, 7.403); using the trim-fill method, the imputed point estimate was 2.668 (0.638, 4.698).

The overall effect sizes on serum and plasma TNF-α levels reported in the forest plot appeared valid, with trivial publication bias effect based on fixed-effects model, because the observed estimates were similar to the adjusted estimates. In contrast, the overall effect sizes on serum and plasma TNF-α levels reported in the forest plot appeared invalid, with significant publication bias effect based on random-effects model, because the observed estimates had high difference with the adjusted estimates.

## 4. Discussion

The present meta-analysis with 55 studies evaluated the serum TNF-α levels (29 studies of adults and 4 studies of children) and plasma TNF-α levels (17 studies of adults and 5 studies of children) in individuals with OSAS, compared to controls. The results showed that plasma and serum TNF-α levels in adult individuals with OSAS were significantly higher than the corresponding levels of control. For children with OSAS serum TNF-α levels, but not plasma TNF-α levels, were statistically significantly higher than those of controls. The present results have clinical importance because elevated serum and plasma TNF-α levels can be a risk factor for the development of further systemic diseases in adults with OSAS. Further, more generally, inflammation may play a crucial role in the pathogenesis of OSAS, and there is evidence that individuals with OSAS have elevated interleukin TNF-α [16,29,40,41,45,48,62,78]. These conclusions also have practical importance because, in addition to routine checks on sleep and sleep-disordered breathing, both children and adults with OSAS need a thorough monitoring of their immune systems.

Elevated TNF-α levels have been associated with the pathophysiology of reoxygenation injury, myocarditis, cardiac allograft vasculopathy, heart failure progression [81], arthritis, diabetes, Crohn’s disease, and also cachexia which correlated with terminal malignancy and AIDS [82]. Further, TNF-α is a well-known inflammatory marker, besides being related to atherosclerosis in males [83]. It is also confirmed that hypoxia raises the expression of proinflammatory molecules such TNF-α [84,85].

The present pattern of results also sheds some light on further new findings: More specifically, for adults, it turned out that compared to healthy controls, TNF-α levels were higher both in blood serum and blood plasma. It follows that at least for adults with OSAS, assessing either blood plasma or blood serum does not appear to be of clinical importance. In contrast, for pediatric samples and compared to healthy controls, higher TNF-α levels were observed for blood serum but not for blood plasma. The quality of the data at hand does not allow a deeper understanding of the underlying physiological mechanisms. Thus, for want of such data, the following admittedly speculative assumptions are advanced. First, it is conceivable that for pediatric samples, the number of four studies on plasma TNF-α was too small, and therefore, the statistical variance of plasma TNF-α was blurred by sample size inconsistencies. Similarly, second, at least among adults, Hirotsu et al. [70] showed that OSAS and inflammatory markers such as IL-6 and TNF-α were associated with gender in a very complex fashion. A closer inspection of the four studies on plasma TNF-α among pediatric samples showed that gender was not introduced as a specific factor or confounder. Given this lack of analyses in these publications, possible gender effects could neither be confirmed, nor declined. Third, Imani et al. [86] showed that for pediatric samples, the definition of OSAS was not applied as strict and consistent as among adult samples. It follows that the zero-differences of plasma TNF-α between children with and without OSAS might mirror inconsistencies as regards the definition of control samples. Fourth, Cameron et al. [87] observed that besides the disease status, in pediatric samples with Inflammatory Bowel Disease, the outcome of an antitumor necrosis factor (TNF) therapy depended from participants’ pubertal stage. Given this, it is conceivable that the zero-difference of plasma TNF-α between pediatric samples with and without OSAS might have been blurred by the pubertal stage. In the same vein, fifth, it is conceivable that pubertal stage and its implicit growth spurt in height and weight might have obscured a clear-cut pattern of results as regards TNF-α levels between children with and without OSAS. Last, even if highly unlikely, one cannot rule out that the pattern of results is an accidental finding.

The pattern of results described in the present meta-analysis and systematic review consistently showed that higher TNF-α levels were associated with pathologically higher BMI scores as a proxy of obesity. Given that above all childhood overweight and childhood obesity is increasing worldwide and is a major concern of public health [88], the association between overweight/obesity, TNF-α levels and OSAS demand particular attention. Following Popa et al. [13], TNF-alpha appeared to be of upmost importance in the development and maintenance of metabolic diseases in which a shift toward a proatherogenic lipid profile and impaired glucose tolerance appeared to occur. Further, investigations assessing the impact of anti-TNF agents on intermediary metabolism seemed to show that a TNF-alpha blockade may improve insulin resistance and lipid profiles, which in turn appeared to be associated with overweight and obesity. Likewise, Ciftci et al. [31] showed in their study that compared to obese males with no OSAS, in obese males with OSAS TNF-alpha levels were significantly higher. Ciftci et al. [31] concluded from their results, that the association between cardiovascular morbidity and OSAS appeared to be best described by the coexistence of other cardiovascular risk factors such as circulating IL-6 and TNF-alpha levels. In the same vein, Sahlman et al. [47] showed in their study that pro-inflammatory markers such as TNF-alpha were markedly increased in patients with mild OSAS. To find a physiological explanation for such processes, Steiropoulos et al. [48] summarized in their introduction that besides the role of energy depot adipose tissue is also an active endocrine organ, releasing proinflammatory cytokines such as Tumor Necrosis Factor-α (TNF-α) and Interleukin-6 (IL-6) which modulate blood pressure, and lipid- and glucose-metabolism. Further, the proinflammatory transcription factor NF-κB appears to be upregulated in OSAS as a result of alterations between hypoxia and reoxygenation and as a result of sleep deprivation. Additionally, NF-κB regulates the expression of inflammatory genes. Given these physiological processes, it appears plausible that OSAS, obesity and poor sleep appear to be highly interrelated.

The results of one study [70] showed that IL-6, triglycerides, and AHI were positively associated with TNF-α, while sex, ghrelin and total cholesterol had a negative association. Further, postmenopausal women had higher TNF-α levels than those of premenopausal women [70]. One study showed that higher TNF-α levels were significantly related to some neurocognitive deficits in children with OSAS [14].

Next, TNF-α appears to mediate both a somnogenic activity and fatigue related to excessive daytime somnolence in obese patients with OSAS [14,28,30]. TNF-α levels were related to the severity of OSAS [31]. One study reported that BMI was weakly related to TNF-α levels [32]. However, Kanbay et al. [41] showed that in individuals with OSAS and obesity serum TNF-α levels were significantly higher than those without obesity. Not surprisingly, there was also a positive albeit modest correlation (r: 0.181; *p*: 0.034) between higher BMI scores and higher TNF-α levels [16].

As regards AHI and TNF-α levels, Kanbay et al. [41] and others [16,78] reported a significant positive correlation between a higher AHI and higher serum TNF-α levels.

Next, Minoguchi et al. [16] reported in their meta-analysis higher plasma and serum TNF-α levels in adults, compared to non-adults.

Despite the novelty of the results, the following limitations should be considered: (1) In all studies, results have not been adjusted for possible confounding factors such as obesity, smoking, or alcohol consumption. (2) The results of the funnel plots showed a publication bias across the studies; it follows that a systematic bias in the data presentation cannot be ruled out. (3) Studies with a small sample size (less than100 cases) had an inadequate power to detect possibly meaningful associations. (4) There was a high heterogeneity among studies in some analyses. (5) Studies reported different cut-off AHI values, which made comparisons between the studies difficult. (6) In some studies, TNF-α levels were considered as secondary outcome. (7) In some studies, the existence of mixed ethnicities might have blurred the associations between the ethnicity and TNF-α levels.

By contrast, the strengths of the meta-analysis were as follows: (1) There were sufficient studies to allow the subgroup analyses. (2) The sensitivity analysis showed a stability of the results. (3) The studies written in other languages than English were included in the meta-analysis.

## 5. Conclusions

The result of the meta-analysis and meta-regression confirmed that compared to healthy controls, individuals with OSAS (adults) had significant higher serum/plasma TNF-α levels, whereas for children with OSAS, this pattern of results was observed for serum TNF-α levels, but not for plasma TNF-α levels. Last, the results of the present meta-analysis showed that elevated TNF-α levels in individuals with OSAS appeared to be related to the severity of the disease. Future studies might investigate if and to what extent interventions on OSAS (e.g., using CPAP devices) favorably impact on TNF-α levels and possibly also on weight regulation.

## Figures and Tables

**Figure 1 life-10-00087-f001:**
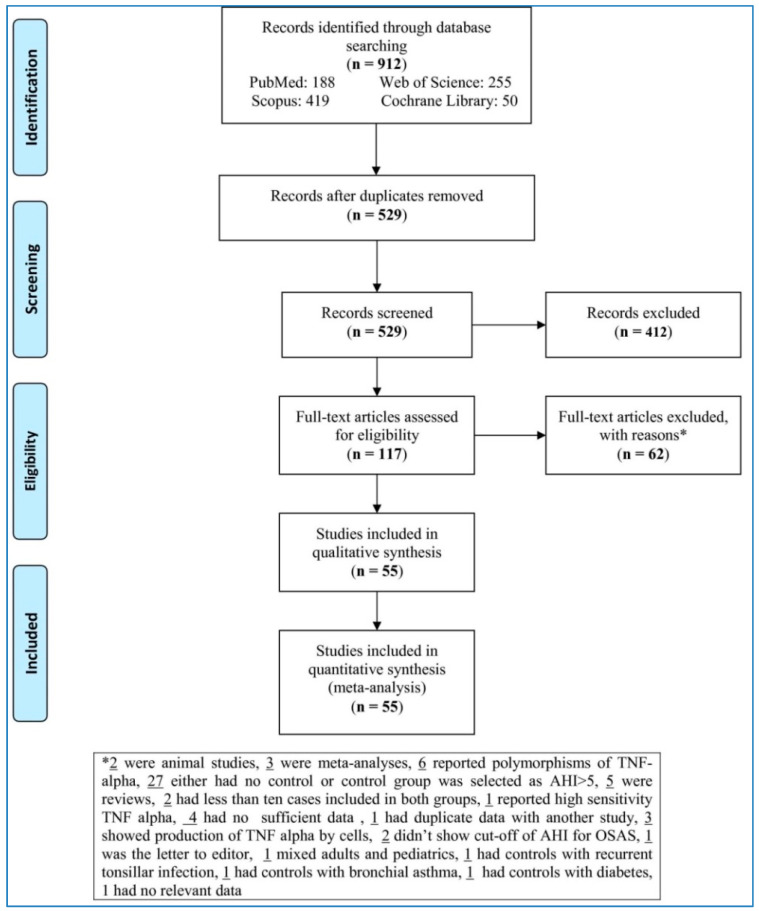
Flowchart of the study selection.

**Figure 2 life-10-00087-f002:**
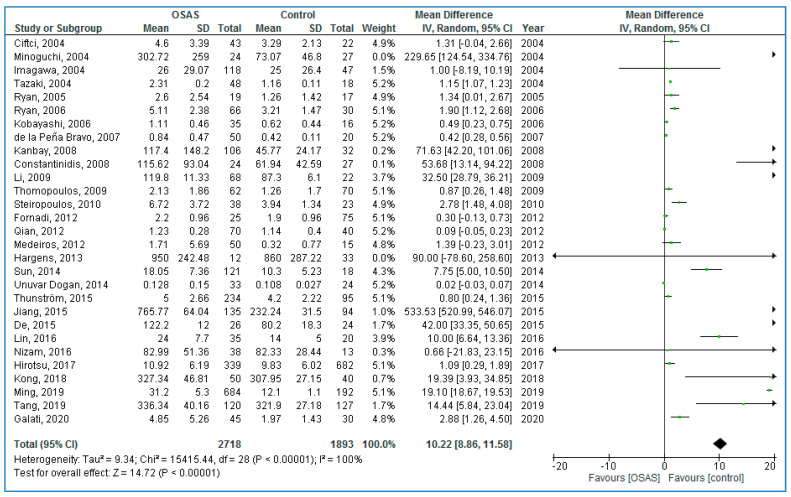
Forest plot of serum tumor necrosis factor alpha levels in adult participants with obstructive sleep apnea syndrome patients compared to controls.

**Figure 3 life-10-00087-f003:**
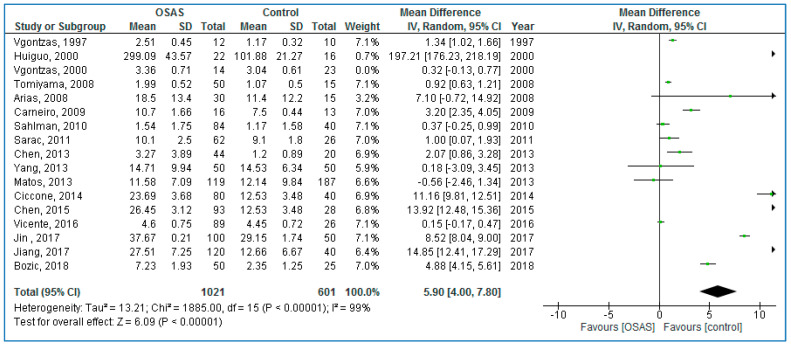
Forest plot of plasma tumor necrosis factor alpha levels in adult participants with obstructive sleep apnea syndrome patients compared to controls.

**Figure 4 life-10-00087-f004:**
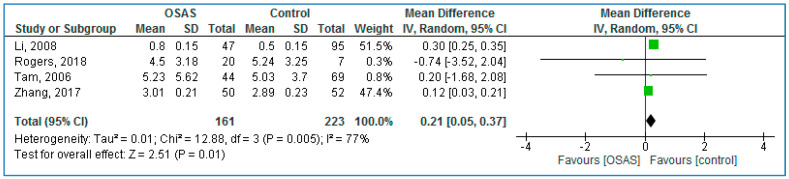
Forest plot of serum tumor necrosis factor alpha levels in pediatric participants with obstructive sleep apnea syndrome patients compared to controls.

**Figure 5 life-10-00087-f005:**
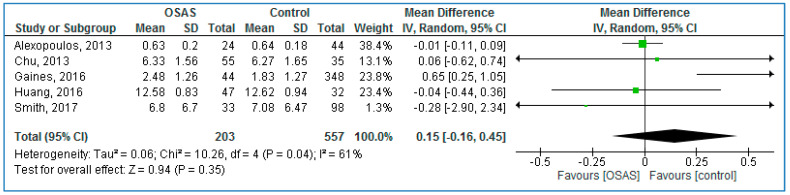
Forest plot of plasma tumor necrosis factor alpha levels in pediatric participants with obstructive sleep apnea syndrome patients compared to controls.

**Figure 6 life-10-00087-f006:**
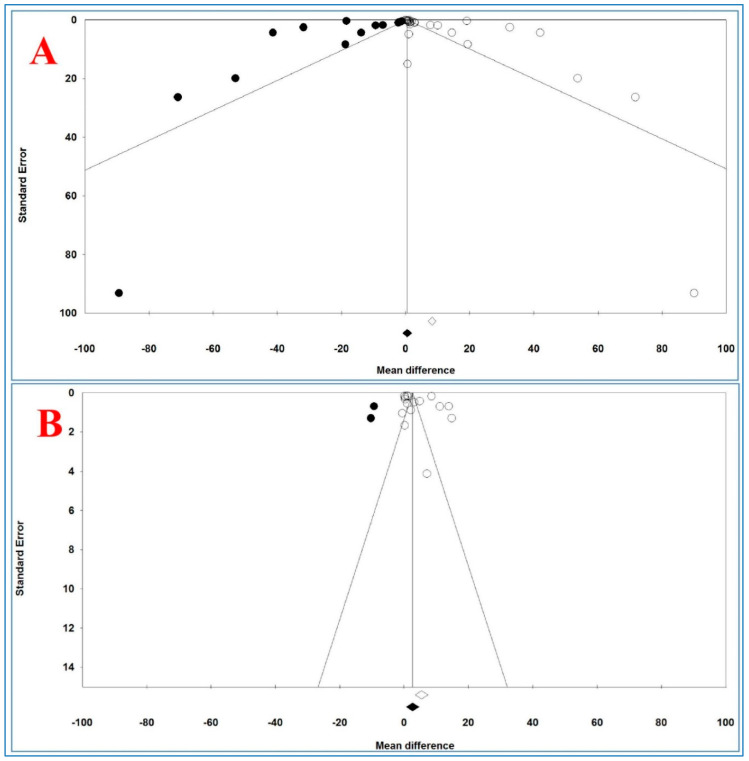
Funnel plot of analysis of tumor necrosis factor alpha levels in adult participants on (**A**) serum and (**B**) plasma in adult participants. Open circles represent observed studies. Black circles represent imputed studies. Open diamond represents the pooled effects from the original studies. Black diamond represents the pooled effects incorporating the imputed studies.

**Table 1 life-10-00087-t001:** Characteristics of studies included in the meta-analysis (*n* = 55).

First Author, Year	Country	Ethnicity	No. of OSAS/Control	OSAS Patients			Controls			Sample
				Age (Years)	BMI (kg/m^2^)	AHI (Events/h)	Age (Years)	BMI (kg/m^2^)	AHI (Events/h)	
Adults										
Vgontzas, 1997 [28]	USA	Mixed	12/10	40.9 ± 2.2	40.5 ± 3.2	63.7 ± 10.3	24.1 ± 0.8	24.6 ± 0.7	0 ± 0	Plasma
Huiguo, 2000 [29]	China	Asian	22/16	47.4 ± 13.6	27.6 ± 3.3	44.0 ± 21.0	47.6 ± 14.7	23.1 ± 3	4.29 ± 2.16	Plasma
Vgontzas, 2000 [30]	USA	Mixed	14/23	46.6 ± 3.0	38.4 ± 1.6	48.7 ± 5.6	43.6 ± 2.5	30.7 ± 1.56	0.88 ± 0.4	Plasma
Ciftci, 2004 [31]	Turkey	Caucasian	43/22	49.6 ± 9.1	31.9 ± 4.1	33.2 ± 5.0	47.2 ± 10.3	31.03 ± 3.1	1.55 ± 0.96	Serum
Imagawa, 2004 [32]	Japan	Asian	118/47	Adult	27.7 ± 4.4	≥5	Adult	22.9 ± 2.9	<5	Serum
Minoguchi, 2004 [16]	Japan	Asian	24/27	50.1 ± 11.7	29.1 ± 2.2	34.1 ± 14.7	48.11 ± 10.5	25.3 ± 1.25	2.27 ± 1.95	Serum
Tazaki, 2004 [33]	Japan	Asian	48/18	50.6 ± 4.8	28.9 ± 1.6	36.05 ± 1.75	48.2 ± 3.0	27.8 ± 0.8	3.7 ± 0.4	Serum
Ryan, 2005 [34]	Ireland	Caucasian	19/17	39.5 ± 20	32.32 ± 16.07	49.6 ± 25.25	39.5 ± 19.5	31.05 ± 15.55	1.03 ± 0.52	Serum
Kobayashi, 2006 [35]	Japan	Asian	35/16	51.4 ± 13.1	27.9 ± 3.6	52.26 ± 14.76	41 ± 13.1	27.4 ± 3.7	<5	Serum
Ryan, 2006 [36]	Ireland	Caucasian	66/30	42.5 ± 8.5	32.5 ± 4.8	35.0 ± 13.9	41 ± 8	30.7 ± 3.1	1.2 ± 1.0	Serum
de la Peña Bravo, 2007 [38]	USA	Mixed	50/20	51.7 ± 1.9	33.2 ± 1.7	51.3 ± 4.2	47.4 ± 1.2	28.4 ± 0.6	2.5 ± 0.5	Serum
Arias, 2008 [39]	Spain	Caucasian	30/15	52.0 ± 13.0	30.5 ± 4.0	43.8 ± 27.0	48.0 ± 10.0	28.7 ± 4.7	3.7 ± 3.3	Plasma
Constantinidis, 2008 [40]	Greece	Caucasian	24/27	45.1 ± 8.2	≥25	23.3 ± 3.6	45.1 ± 8.2	≥25	3.5 ± 0.4	Serum
Kanbay, 2008 [41]	Turkey	Caucasian	106/32	51.39 ± 10.37	31.06 ± 5.87	40.14 ± 14.30	44.79 ± 13.35	28.25 ± 5.49	1.96 ± 1.08	Serum
Tomiyama, 2008 [43]	Japan	Asian	50/15	51.4 ± 13.0	26.9 ± 4.2	42.7 ± 27.9	53.0 ± 10.0	24.3 ± 2.5	<5	Plasma
Carneiro, 2009 [44]	Brazil	Mixed	16/13	40.1 ± 2.8	46.9 ± 2.0	65.7 ± 9.9	38.8 ± 3.3	42.8 ± 1.3	3.2 ± 0.5	Plasma
Li, 2009 [45]	China	Asian	68/22	45.3 ± 11.1	20.7 ± 10.1	38.9 ± 26.5	43.0 ± 93.0	23.3 ± 2.0	2.9 ± 1.3	Serum
Thomopoulos, 2009 [46]	Greece	Caucasian	62/70	48.1 ± 7.6	31.9 ± 4.9	31.6 ± 2.0	48.1 ± 3.9	32.1 ± 3.0	0.4 ± 4.0	Serum
Sahlman, 2010 [47]	Finland	Caucasian	84/40	50.4 ± 9.3	32.5 ± 3.3	9.6 ± 2.9	45.6 ± 11.5	31.5 ± 3.5	1.9 ± 1.4	Plasma
Steiropoulos, 2010 [48]	Greece	Caucasian	38/23	45.5 ± 10.5	36.4 ± 7.4	61.0 ± 27.0	43.7 ± 6.7	34.5 ± 3.7	5.3 ± 3.2	Serum
Sarac, 2011 [49]	Turkey	Caucasian	62/26	50.0 ± 19.7	33.7 ± 4.2	29.5 ± 1.9	49.7 ± 11.1	34.3 ± 5.4	<5	Plasma
Fornadi, 2012 [50]	Canada	Mixed	25/75	54.0 ± 12.0	29.0 ± 5.0	≥5	50.0 ± 13.0	26.0 ± 5.0	<5	Serum
Medeiros, 2012 [51]	Brazil	Mixed	50/15	62.3 ± 7.8	25.5 ± 4.0	≥5	62.50 ± 8.4	25.8 ± 4.0	<5	Serum
Qian, 2012 [52]	China	Asian	70/40	45.8 ± 8.2	28.9 ± 2.3	≥5	46.3 ± 8.1	24.1 ± 2.3	<5	Serum
Chen, 2013 [54]	China	Asian	44/20	21.4 ± 11.95	27.12 ± 3.53	14.56 ± 2.85	42 ± 11	26.0 ± 3.3	3.3 ± 0.9	Plasma
Hargens, 2013 [56]	USA	Mixed	12/33	22.8 ± 0.8	32.4 ± 1.0	25.4 ± 5.4	21.9 ± 0.6	29.3 ± 0.5	2.1 ± 0.3	Serum
Matos, 2013 [57]	Brazil	Mixed	119/187	51.2	29.7	≥5	40.7	24.5	<5	Plasma
Yang, 2013 [58]	China	Asian	50/25	53.5 ± 6.9	27.4 ± 2.9	24.5 ± 15.9	53.0 ± 7.0	26.27 ± 1.9	3.0 ± 1.0	Plasma
Ciccone, 2014 [59]	Italy	Caucasian	80/40	52.8 ± 10.6	28.6 ± 3.0	33.9 ± 21	52.3 ± 10.5	28.2 ± 2.7	2.1 ± 1.1	Plasma
Sun, 2014 [60]	China	Asian	121/18	43.3 ± 11.6	27.1 ± 3.1	40.8 ± 10.9	43.9 ± 13.4	25.7 ± 3.8	2.1 ± 1.8	Serum
Unuvar Dogan, 2014 [61]	Turkey	Caucasian	33/24	45.3 ± 8.5	31.0 ± 1.7	47.2 ± 23.2	40.5 ± 9.5	30.7 ± 1.5	3.6 ± 1.8	Serum
Chen, 2015 [62]	China	Asian	93/28	42.28 ± 8.55	28.84 ± 3.82	27 ± 4.06	43.7 ± 9.8	26.4 ± 2.5	2.6 ± 1.2	Plasma
De, 2015 [63]	Italy	Caucasian	26/24	41.8 ± 7.4	33.0 ± 5.2	26.15 ± 12.1	43.7 ± 8.2	30.8 ± 4.3	1.65 ± 0.9	Serum
Jiang, 2015 [64]	China	Asian	135/94	48.7 ± 12.1	27.48 ± 2.56	24.57 ± 15.9	47.2 ± 13.5	27.52 ± 2.58	1.60 ± 1.61	Serum
Thunström, 2015 [65]	Sweden	Caucasian	234/95	65.3 ± 7.1	26.8 ± 2.1	28.9 ± 13.7	61.4 ± 9.5	25.2 ± 2.5	3.1 ± 1.3	Serum
Lin, 2016 [67]	Taiwan	Asian	35/20	46 ± 7	29.2 ± 1.9	59.3 ± 23.2	43 ± 8	28.2 ± 1.6	3.6 ± 0.8	Serum
Nizam, 2016 [68]	Turkey	Caucasian	39/13	47.3 ± 10.4	33.2 ± 56.4	45.6 ± 20.7	43.23 ± 9.08	31.71 ± 4.56	2.64 ± 1.82	Serum
Vicente, 2016 [69]	Spain	Caucasian	89/26	45.33 ± 14.81	30.03 ± 5.04	28 ± 23.70	45 ± 11.11	28.7 ± 4.37	1.9 ± 2.7	Plasma
Hirotsu, 2017 [70]	Brazil	Mixed	339/682	50.8 ± 13.2	29.6 ± 5.8	19.3 ± 9.44	38.2 ± 12.7	25.4 ± 3.8	2.5 ± 10.4	Serum
Jiang, 2017 [71]	China	Asian	120/40	24.82 ± 10.7	28.5 ± 5.13	46.6 ± 4.56	46.5 ± 12.3	27.5 ± 6.2	2.13 ± 1.26	Plasma
Jin, 2017 [72]	China	Asian	100/50	55.28 ± 7.13	26.75 ± 3.50	38.01 ± 8.04	56.13 ± 6.21	25.19 ± 2.45	3.62 ± 1.54	Plasma
Bozic, 2018 [75]	Croatia	Caucasian	50/25	53.0 ± 11.9	28.9 ± 2.7	35.0 ± 11.0	52.5 ± 10.2	27.8 ± 2.2	<5	Plasma
Kong, 2018 [76]	China	Asian	50/40	54.34 ± 14.38	26.86 ± 3.12	37.34 ± 19.02	50.42 ± 8.35	22.26 ± 3.54	3.31 ± 1.09	Serum
Ming, 2019 [78]	China	Asian	684/192	51.34 ± 5.16	≤30	31.15 ± 9.12	52.18 ± 4.51	≤30	4.34 ± 2.01	Serum
Tang, 2019 [79]	China	Asian	120/127	48.88 ± 9.76	26.86 ± 3.12	39.00 ± 18.38	47.37 ± 9.12	22.46 ± 3.29	3.31 ± 1.09	Serum
Galati, 2020 [80]	Italy	Caucasian	45/30	53.9 ± 11.6	28 ± 2.2	≥5	55 ± 5.8	26.3 ± 1.8	<5	Serum
Pediatrics										
Tam, 2006 [37]	Australia	Mixed	44/69	7.3 ± 3.7	19.4 ± 5.5	5.3 ± 6.5	7.6 ± 4.0	17.9 ± 3.9	0 ± 0	Serum
Li, 2008 [42]	China	Asian	47/95	11.1 ± 1.27	NR	14.1 ± 8.0	10.7 ± 1.3	NR	0.7 ± 0.6	Serum
Alexopoulos, 2013 [53]	Greece	Caucasian	24/44	5.7 ± 2	<25	11.5 ± 5.1	6.4 ± 2.3	<25	1.3 ± 0.65	Plasma
Chu, 2013 [55]	China	Asian	55/35	Child	NR	≥5	Child	NR	<5	Plasma
Gaines, 2016 [66]	USA	Mixed	44/348	18.41 ± 2.2	NR	12.08 ± 4.64	18.4 ± 2.2	NR	1.65 ± 4.81	Plasma
Huang, 2016 [14]	Taiwan	Asian	47/32	7.84 ± 0.56	16.95 ± 0.47	9.13 ± 1.67	7.02 ± 0.65	6.55 ± 0.58	0.41 ± 0.07	Plasma
Smith, 2017 [73]	USA	Mixed	33/98	9.0 ± 2.7	23.4 ± 13.5	15.2 ± 11.7	19.66 ± 4.44	9.68 ± 2.5	1.03 ± 0.46	Plasma
Zhang, 2017 [74]	China	Asian	50/52	6.6	NR	≥5	6.4	NR	<5	Serum
Rogers, 2018 [77]	USA	Mixed	20/7	7.9 ± 2.2	<25	13.1 ± 9.8	7.9 ± 2.2	<25	0.8 ± 0.3	Serum

NR, Not reported; OSAS, Obstructive sleep apnea syndrome; AHI, Apnea-hypopnea index; BMI, Body mass index.

**Table 2 life-10-00087-t002:** Subgroup analysis on serum and plasma levels of tumor necrosis factor alpha in adult participants.

Subgroup Analysis of Serum Level (*n*)	MD (95%CI), *p*-Value, I^2^ (%), P_h_	Subgroup Analysis of Plasma Level (*n*)	MD (95%CI), *p*-Value, I^2^ (%), P_h_
Overall (29)	10.22 (8.86, 11.58), <0.00001; 100, <0.00001	Overall (17)	5.41 (3.66, 7.15), <0.00001, 99, <0.00001
Ethnicity	-	Ethnicity	-
Caucasian (12)	2.20 (1.06, 3.34),0.0002, 94, <0.00001	Caucasian (6)	3.72 (1.04, 6.40), 0.007, 99, <0.00001
Asian (12)	37.11 (33.27, 40.95), <0.00001, 100, <0.00001	Asian (7)	14.92 (9.86, 19.98), <0.00001, 100, <0.00001
Mixed (5)	0.43 (0.30, 0.56), <0.00001, 26, 0.25	Mixed (4)	1.22 (0.10, 2.34), 0.03, 93, <0.00001
Mean BMI of OSAS patients, kg/m^2^	-	Mean BMI of OSAS patients, kg/m^2^	-
>30 (11)	1.27 (0.67, 1.86), <0.0001, 95, <0.000001	>30 (7)	1.06 (0.34, 1.78), 0.004, 91, <0.00001
≤30 (17)	23.07 (20.18, 25.96), <0.00001, 100, <0.00001	≤30 (10)	10.82 (7.14, 14.49), <0.00001, 99, <0.00001
Mean BMI of controls, kg/m^2^	-	Mean BMI of controls, kg/m^2^	-
>30 (8)	2.28 (0.86, 3.70), 0.002, 95, <0.00001	>30 (4)	1.19 (0.00, 2.38), 0.05, 92, <0.00001
≤30 (20)	18.64 (16.33, 20.95), <0.00001, 100, <0.00001	≤30 (13)	7.32 (5.02, 9.62), <0.00001, 99, <0.00001
Total number of participants	-	Total number of participants	-
>100 (10)	55.47 (47.21, 63.72), <0.00001, 100, <0.00001	>100 (7)	6.91 (2.20, 11.63), 0.004, 100, <0.00001
≤100 (19)	2.15 (1.57, 2.72), <0.00001, 98, <0.00001	≤100 (10)	2.87 (1.37, 4.37), 0.0002, 98, <0.00001
Mean AHI of OSAS patients, events/h	-	Mean AHI of OSAS patients, events/h	-
>30 (18)	5.72 (4.32, 7.12), <0.00001, 100, <0.00001	>30 (10)	8.15 (5.59, 10.71), <0.00001, 99, <0.00001
≤30 (6)	127.01 (100.56, 153.47), <0.00001, 100, <0.00001	≤30 (6)	2.97 (0.21, 5.73), 0.03, 99, <0.00001

BMI, Body mass index; CI, Confidence interval; OSAS, Obstructive sleep apnea syndrome; MD, Mean difference; P_h_, P_heterogeneity._ Bold numbers show statistically significant value (*p*-value < 0.05).

**Table 3 life-10-00087-t003:** Subgroup analysis on serum and plasma levels of tumor necrosis factor alpha in adult Caucasian participants.

Subgroup Analysis of Serum Level (*n*)	MD (95%CI), *p*-Value, I^2^ (%), P_h_	Subgroup Analysis of Plasma Level (*n*)	MD (95%CI), *p*-Value, I^2^ (%), P_h_
Overall (12)	2.20 (1.06, 3.34), 0.0002, 94, <0.00001	Overall (6)	3.72 (1.04, 6.40), 0.007, 99, <0.00001
Mean BMI of OSAS patients, kg/m^2^	-	Mean BMI of OSAS patients, kg/m^2^	-
>30 (9)	2.58 (1.07, 4.10), 0.0008, 95, <0.00001	>30 (4)	0.43 (−0.010, 0.96), 0.11, 50, 0.11
≤30 (2)	1.70 (−0.32, 3.71), 0.10, 82, 0.02	≤30 (2)	7.99 (1.84, 14.15), 0.01, 98, <0.00001
Mean BMI of controls, kg/m^2^	-	Mean BMI of controls, kg/m^2^	-
>30 (8)	2.28 (0.86, 3.70), 0.002, 95, <0.00001	>30 (2)	0.56 (0.05, 1.08), 0.03, 18, 27
≤30 (3)	3.29 (−1.09, 7.67), 0.14, 93, <0.00001	≤30 (4)	5.64 (0.77, 10.52), 0.02, 99, <0.00001
Total number of participants	-	Total number of participants	-
>100 (3)	1.13 (−0.78, 3.04), 0.25, 91, <0.0001	>100 (3)	3.82 (−0.22, 7.85), 0.06, 99, <0.00001
≤100 (9)	3.21 (1.37, 5.05), 0.0006, 95, <0.00001	≤100 (3)	3.52 (0.03, 7.02), 0.05, 95, <0.00001
Mean AHI of OSAS patients, events/h	-	Mean AHI of OSAS patients, events/h	-
>30 (8)	1.40 (0.34, 2.46), 0.010, 91, <0.00001	>30 (3)	7.80 (2.47, 13.13), 0.004, 97, <0.00001
≤30 (2)	23.20 (−28.02, 74.41), 0.37, 85, 0.01	≤30 (3)	0.26 (−0.01, 0.53), 0.06, 34, 0.22

BMI, Body mass index; CI, Confidence interval; OSAS, Obstructive sleep apnea syndrome; MD, Mean difference; P_h_, P_heterogeneity._ Bold numbers show statistically significant value (*p*-value < 0.05).

**Table 4 life-10-00087-t004:** Subgroup analysis on serum and plasma levels of tumor necrosis factor alpha in adult Asian participants.

Subgroup Analysis of Serum Level (*n*)	MD (95%CI), *p*-Value, I^2^ (%), P_h_	Subgroup Analysis of Plasma Level (*n*)	MD (95%CI), *p*-Value, I^2^ (%), P_h_
Overall (12)	37.11 (33.27, 40.95), <0.00001, 100, <0.00001	Overall (7)	14.92 (9.86, 19.98), <0.00001, 100, <0.00001
Mean BMI of OSAS patients, kg/m^2^	-	Mean BMI of OSAS patients, kg/m^2^	-
>30 (0)	-	>30 (0)	-
≤30 (12)	37.11 (33.27, 40.95), <0.00001, 100, <0.00001	≤30 (7)	14.92 (9.86, 19.98), <0.00001, 100, <0.00001
Mean BMI of controls, kg/m^2^	-	Mean BMI of controls, kg/m^2^	-
>30 (0)	-	>30 (0)	-
≤30 (12)	37.11 (33.27, 40.95), <0.00001, 100, <0.00001	≤30 (7)	14.92 (9.86, 19.98), <0.00001, 100, <0.00001
Total number of participants	-	Total number of participants	-
>100 (6)	89.88 (74.59, 105.17), <0.00001, 100, <0.00001	>100 (3)	12.34 (7.84, 16.84), <0.00001, 97, <0.00001
≤100 (6)	7.80 (5.76, 9.85), <0.00001, 99, <0.00001	≤100 (4)	21.51 (13.15, 29.87), <0.00001, 99, <0.00001
Mean AHI of OSAS patients, events/h	-	Mean AHI of OSAS patients, events/h	-
>30 (9)	12.94 (7.74, 18.13), <0.00001, 100, <0.00001	>30 (4)	12.48 (17.04, 31.52), <0.00001, 100, <0.00001
≤30 (0)	-	≤30 (3)	5.45 (−3.55, 14.45), 0.24, 99, <0.00001

BMI, Body mass index; CI, Confidence interval; OSAS, Obstructive sleep apnea syndrome; MD, Mean difference; P_h_, P_heterogeneity._ Bold numbers show statistically significant value (*p*-value < 0.05).

**Table 5 life-10-00087-t005:** Subgroup analysis on serum and plasma levels of tumor necrosis factor alpha in adult participants including mean BMI of OSAS patients >30 kg/m^2.^

Subgroup Analysis of Serum Level (*n*)	MD (95%CI), *p*-Value, I^2^ (%), P_h_	Subgroup Analysis of Plasma Level (*n*)	MD (95%CI), *p*-Value, I^2^ (%), P_h_
Overall (11)	1.27 (0.67, 1.86), <0.0001, 95, <0.000001	Overall (7)	1.06 (0.34, 1.78), 0.004, 91, <0.00001
Mean BMI of controls, kg/m^2^	-	Mean BMI of controls, kg/m^2^	-
>30 (8)	2.28 (0.86, 3.70), 0.002, 95, <0.00001	>30 (4)	1.19 (0.00, 2.38), 0.05, 92, <0.00001
≤30 (3)	40.64 (−24.23, 105.51), 0.22, 92, <0.00001	≤30 (3)	0.88 (−0.30, 2.07), 0.14, 93, <0.00001
Total number of participants	-	Total number of participants	-
>100 (2)	34.66 (−34.62, 103.93), 0.33, 95, <0.00001	>100 (2)	0.20 (−0.09, 0.48), 0.17, 0, 0.53
≤100 (9)	1.27 (0.65, 1.89), <0.0001, 95, <0.00001	≤100 (5)	1.51 (0.53, 2.49), 0.003, 90, <0.00001
Mean AHI of OSAS patients, events/h	-	Mean AHI of OSAS patients, events/h	-
>30 (9)	0.96 (0.50, 1.42), <0.0001, 92, <0.00001	>30 (4)	1.69 (0.47, 2.90), 0.007, 92, <0.00001
≤30 (2)	42.13 (33.48, 50.77), <0.00001, 0, 0.58	≤30 (3)	0.26 (−0.01, 0.53), 0.06, 34, 0.22

BMI, Body mass index; CI, Confidence interval; OR, OSAS, Obstructive sleep apnea syndrome; MD, Mean difference; P_h_, P_heterogeneity._ Bold numbers show statistically significant value (*p*-value < 0.05).

**Table 6 life-10-00087-t006:** Subgroup analysis on serum and plasma levels of tumor necrosis factor alpha in adult participants including mean BMI of OSAS patients ≤30 kg/m^2.^

Subgroup Analysis of Serum Level (*n*)	MD (95%CI), *p*-Value, I^2^ (%), P_h_	Subgroup Analysis of Plasma Level (*n*)	MD (95%CI), *p*-Value, I^2^ (%), P_h_
Overall (17)	23.07 (20.18, 4.50), <0.00001, 100, <0.00001	Overall (10)	10.82 (7.14, 14.49), <0.00001, 99, <0.00001
Mean BMI of controls, kg/m^2^	-	Mean BMI of controls, kg/m^2^	-
>30 (0)	-	>30 (0)	-
≤30 (17)	23.07 (20.18, 4.50), <0.00001, 100, <0.00001	≤30 (10)	10.82 (7.14, 14.49), <0.00001, 99, <0.00001
Total number of participants	-	Total number of participants	-
>100 (8)	63.51 (53.65, 73.45), <0.00001, 100, <0.00001	>100 (6)	8.12 (4.46, 11.79), <0.0001, 98, <0.00001
≤100 (9)	4.42 (3.10, 5.74), <0.00001, 98, <0.00001	≤100 (4)	13.79 (8.24, 19.35), <0.00001, 99, <0.00001
Mean AHI of OSAS patients, events/h	-	Mean AHI of OSAS patients, events/h	-
>30 (9)	12.94 (7.74, 18.13), <0.00001, 100, <0.00001	>30 (6)	16.13 (11.20, 21.06), <0.00001, 100, <0.00001
≤30 (3)	173.00 (139.66, 206.34), <0.00001, 100, <0.00001	≤30 (3)	5.45 (−3.55, 14.45), 0.24, 99, <0.00001

BMI, Body mass index; CI, Confidence interval; OR, OSAS, Obstructive sleep apnea syndrome; MD, Mean difference; NA, Not available; P_h_, P_heterogeneity._ Bold numbers show statistically significant value (*p*-value < 0.05).

**Table 7 life-10-00087-t007:** Meta-regression analysis based on some variables for serum and plasma levels of tumor necrosis factor alpha in obstructive sleep apnea syndrome patients compared with controls in adult participants.

Year of Publication	R	Adjusted R^2^	*p*	Mean Age of OSAS Patients	R	Adjusted R^2^	*p*	Mean Age of Controls	R	Adjusted R^2^	*p*
Serum	0.014	−0.036	0.942	Serum	0.111	−0.024	0.567	Serum	0.081	−0.030	0.678
Plasma	0.384	0.091	0.128	Plasma	0.018	−0.066	0.946	Plasma	0.079	−0.060	0.764
Mean BMI of OSAS patients	R	Adjusted R^2^	*p*	Mean BMI of controls	R	Adjusted R^2^	*p*	Mean AHI of OSAS patients	R	Adjusted R^2^	*p*
Serum	0.122	−0.025	0.545	Serum	0.159	−0.014	0.429	Serum	0.339	0.075	0.105
Plasma	0.201	−0.024	0.439	Plasma	0.292	0.024	0.256	Plasma	0.126	−0.054	0.641
Number of participants	R	Adjusted R^2^	*p*	-	-	-	-	-	-	-	-
Serum	0.004	−0.36	0.985	-	-	-	-	-	-	-	
Plasma	0.202	−0.023	0.438	-	-	-	-	-	-	-	-

BMI, Body mass index; OSAS, Obstructive sleep apnea syndrome. R: Correlation coefficient.

**Table 8 life-10-00087-t008:** Quality assessment scores of the studies involved in the meta-analysis.

The First Author, Year	Selection	Comparability	Exposure	Total Points
Adults				
Vgontzas, 1997 [28]	***	-	***	6
Huiguo, 2000 [29]	***	**	***	8
Vgontzas, 2000 [30]	***	**	***	8
Ciftci, 2004 [31]	***	**	***	8
Imagawa, 2004 [32]	***	*	***	7
Minoguchi, 2004 [16]	***	**	***	8
Tazaki, 2004 [33]	***	**	***	8
Ryan, 2005 [34]	***	**	***	8
Kobayashi, 2006 [35]	***	**	***	8
Ryan, 2006 [36]	***	**	***	8
de la Peña Bravo, 2007 [38]	***	**	***	8
Arias, 2008 [39]	***	**	***	8
Constantinidis, 2008 [40]	***	**	***	8
Kanbay, 2008 [41]	***	**	***	8
Tomiyama, 2008 [43]	***	**	***	8
Carneiro, 2009 [44]	***	*	***	7
Li, 2009 [45]	***	**	***	8
Thomopoulos, 2009 [46]	***	**	***	8
Sahlman, 2010 [47]	***	**	***	8
Steiropoulos, 2010 [48]	***	**	***	8
Sarac, 2011 [49]	***	*	***	7
Fornadi, 2012 [50]	***	**	***	8
Medeiros, 2012 [51]	***	**	***	8
Qian, 2012 [52]	***	**	***	8
Chen, 2013 [54]	***	*	***	7
Hargens, 2013 [56]	***	**	***	8
Matos, 2013 [57]	***	**	***	8
Yang, 2013 [58]	***	**	***	8
Ciccone, 2014 [59]	***	**	***	8
Sun, 2014 [60]	***	**	***	8
Unuvar Dogan, 2014 [61]	***	**	***	8
Chen, 2015 [62]	***	**	***	8
De, 2015 [63]	***	**	***	8
Jiang, 2015 [64]	***	**	***	8
Thunström, 2015 [65]	****	**	***	9
Lin, 2016 [67]	***	**	***	8
Nizam, 2016 [68]	***	**	***	8
Vicente, 2016 [69]	***	**	***	8
Hirotsu, 2017 [70]	****	**	***	9
Jiang, 2017 [71]	***	*	***	7
Jin, 2017 [72]	***	**	***	8
Bozic, 2018 [75]	***	**	***	8
Kong, 2018 [76]	***	**	***	8
Ming, 2019 [78]	***	*	***	7
Tang, 2019 [79]	***	**	***	8
Galati, 2020 [80]	****	**	***	9
Children				
Tam, 2006 [37]	****	**	***	9
Li, 2008 [42]	***	*	***	7
Alexopoulos, 2013 [53]	***	*	***	7
Chu, 2013 [55]	***	-	***	6
Gaines, 2016 [66]	***	*	***	7
Huang, 2016 [14]	***	*	***	7
Smith, 2017 [73]	***	*	***	7
Zhang, 2017 [74]	***	*	***	7
Rogers, 2018 [77]	***	*	***	7

Each asterisk (*) denotes 1 point.

**Table 9 life-10-00087-t009:** Sensitivity analysis on the results of serum and plasma levels in adults participants.

First Author, Year	Sample	Reason for Removing	MD (95%CI)	Z	*p*-Value	I^2^	P_h_
Jiang, 2015 [64]	Serum	Outlier data	4.53 (3.48, 5.57)	8.87	<0.00001	100	<0.00001
Huiguo, 2000 [29]	Plasma	Outlier data	4.18 (2.58, 5.78)	5.11	<0.00001	99	<0.00001

CI, Confidence interval; MD, Mean difference; P_h_, P_heterogeneity._

**Table 10 life-10-00087-t010:** The results of trim-and-fill method.

Sample	Value	Studies Trimmed	Fixed-Effects			Random-Effects			Q Value
			Point Estimate	Lower Limit	Upper Limit	Point Estimate	Lower Limit	Upper Limit	
Serum	Observed	-	0.40992	0.36474	0.45509	8.24489	7.00639	9.48339	8710.33950
	Adjusted	15	0.32271	0.27770	0.36771	0.52315	−0.88882	1.93512	17050.9022
Plasma	Observed	-	2.46259	2.32314	2.60204	5.45206	3.50092	7.40320	2508.21501
	Adjusted	3	2.29455	2.15605	2.43306	2.66813	0.63833	4.69794	3161.99635

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
