# Peer review of "Serum and Plasma Tumor Necrosis Factor Alpha Levels in Individuals with Obstructive Sleep Apnea Syndrome: A Meta-Analysis and Meta-Regression"

_life, 2020, doi:10.3390/life10060087_

Round 1
Reviewer 1 Report
This meta-analysis confirmed elevated TNF α in OSA pts. I would suggest a more detailed discussion: (1) about observed differences between serum and plasma levels of TNF (2) the role of obesity and (3) the differences between children and adults.
Author Response
Thank you for your kind efforts; please see the point-by-point-response attached as a separate file.

Reviewer 2 Report
In this manuscript, the authors presented a comprehensive meta-analysis on the TNF-a levels in serum and plasma of individuals with and without OSAS. The workflow is by large acceptable. This work gave a general overview of current knowledge on the association of TNF-a levels with OSAS, and should be very useful to related research fields. There are a few comments and concerns that I hope the authors can consider/specify/clarify before I would recommend it for publication in Life.
(1) "Mixed race" is not a specific "race". Listing and comparing Caucasians and Asian v.s. "mixed race" seems inappropriate. While I acknowledge the limitation of information availability from published studies, the cross analysis between these race-based subgroups could raise concerns and become misleading.
(2) In Figures 2-5, how was "weight" calculated? This is a critical parameter that the results sensitively depended on. Please specify in the method section. The authors should also comment on why it was calculated in this way but not others (e.g. based on number of cases/total cases)
(3) Specify the age cut-off for "adults" and "children", if available.
Author Response

(The authors gave the same response as above.)

Round 2
Reviewer 1 Report
No other comments
Reviewer 2 Report
The authors have addressed all my concerns. The authors should give another round of careful check of the writing to polish and smooth out certain expressions. I recommend the manuscript to be published in the Life journal.